# Decision-making in plants under competition

Michal Gruntman[1], Dorothee Groß[1], Maria Májeková[1,2,3] & Katja Tielbörger[1]

Plants can plastically respond to light competition in three strategies, comprising vertical growth, which promotes competitive dominance; shade tolerance, which maximises performance under shade; or lateral growth, which offers avoidance of competition. Here, we test the hypothesis that plants can 'choose' between these responses, according to their abilities to competitively overcome their neighbours. We study this hypothesis in the clonal plant *Potentilla reptans* using an experimental setup that simulates both the height and density of neighbours, thus presenting plants with different light-competition scenarios. *Potentilla reptans* ramets exhibit the highest vertical growth under simulated short-dense neighbours, highest specific leaf area (leaf area/dry mass) under tall-dense neighbours, and tend to increase total stolon length under tall-sparse neighbours. These responses suggest shifts between 'confrontational' vertical growth, shade tolerance and lateral-avoidance, respectively, and provide evidence that plants adopt one of several alternative plastic responses in a way that optimally corresponds to prevailing light-competition scenarios.

[1] Plant Ecology Group, Institute of Evolution and Ecology, University of Tübingen, Auf der Morgenstelle 5, 72076 Tübingen, Germany. [2] Department of Soil Science, Faculty of Natural Science, Comenius University in Bratislava, Ilkovičova 6, 842 15 Mlynska dolina, Bratislava, Slovak Republic. [3] Department of Botany, Faculty of Science, University of South Bohemia, 370 05 České Budějovice, Czech Republic. Correspondence and requests for materials should be addressed to M.G. (email: michal.gruntman@bot.uni-tuebingen.de)

Competition plays a fundamental role in plant ecology, but in many habitats the occurrence of neighbours can vary over space and time and plants have evolved both the ability to detect the presence of neighbours and to plastically adjust their phenotypes in response to it[1–5]. In particular, competition for light is known to elicit two types of well-studied plastic responses in plants, known as shade avoidance and shade-tolerance[2, 6–8]. Shade-avoidance responses comprise a suite of morphological adjustments, such as enhanced elongation of the stem or petioles, and their increased vertical inclination (i.e. vertical angle). These responses result in vertical growth, thereby allowing plants to position their leaves in favourable high-light conditions[2, 3, 6, 7, 9, 10]. Shade-avoidance responses are elicited in plants via cues indicating neighbour proximity, including the reduction in photosynthetically active radiation (PAR) and in the ratio of red to far-red wavelengths (R:FR)[7, 9, 11–13], as well as mechanical stimulation or plant volatiles[12, 14, 15]. Although termed 'shade avoidance', vertical growth does not necessarily provide avoidance of competitive interactions, particularly in dense stands where plants might engage in a competitive arms race for light[16, 17]. Instead, vertical elongation could be depicted as a 'pre-emptive or confrontational' behaviour by which plants can deprive each other of light and gain a competitive advantage[18].

Shade-tolerance responses consist of several well-studied morphological and physiological adjustments that promote plant performance under limited light conditions[6, 8]. Such tolerance is achieved by increasing light-capture efficiency, e.g. by increasing leaf area at the expense of leaf thickness, which results in increased leaf area to leaf mass ratio (specific leaf area)[6, 8, 19, 20]. Shade tolerance can also be achieved by adjusting photosynthetic responses, e.g. by decreasing photosynthetic capacity[6, 8, 20]. Unlike shade-avoidance responses, which are induced by cues specifically indicating neighbour proximity, shade-tolerance responses are known to be elicited in plants mainly via decreases in PAR levels[8, 19].

In addition, some plants can respond to light-competition cues by employing competition-avoidance behaviours that could improve light intercept and minimise competitive interactions, e.g. by growing away from their neighbours[10, 18, 21]. This response is particularly suitable for procumbent plants, which can grow horizontally[21], but even more so for clonal plants, which can 'move', e.g. by increasing internode length of their stolons or rhizomes, thus actively positioning new ramets in less crowded patches[10, 22–24].

Plants, and clonal plants in particular, can therefore employ three categories of response to light competition: vertical 'confrontational' growth, shade tolerance or lateral avoidance[8, 18, 25, 26]. Previous studies have examined the different evolutionary backgrounds that might select for either one of these responses, such as the canalisation of shade tolerance in tropical woody species[27, 28]. Other studies have shown that in habitats where light competition is extremely strong, such as in forest understories, plants are less responsive to light competition cues in their vertical elongation[29–33]. However, even within the same habitat, the strength of competitive interactions imposed on plants can unpredictably vary in both space and time due to stochastic processes such as dispersal or environmental heterogeneity, and plants can encounter neighbours with different stature, age or density. Plants from such heterogeneous environments would therefore benefit from being able to (a) perceive the competitive ability of their neighbours relative to their own and (b) 'choose' between the three response categories according to the probable fitness outcome of each[18, 26]. Accordingly, Novoplansky[18] proposed that plants are likely to invest in vertical growth when growing among equal-sized competitors, but when challenged by taller competitors that are less likely to be outgrown, plants are expected to shift to either shade tolerance or, in the case of procumbent or clonal plants, to avoidance responses via horizontal spread.

Previous studies have provided evidence that plants can assess the size of their neighbours and adjust their vertical elongation accordingly[34–37]. Additionally, plants can refrain from vertical elongation when facing kin neighbours with which they can cooperate[13], or when confronted with neighbours they are less likely to outcompete, such as trees[23, 38, 39]. If plants can thus evaluate the competitive ability of their neighbours, it is likely that they can not only adopt the optimal strategy within a certain response (e.g. extant of vertical elongation) but also between different responses to competition. Namely, they could shift to alternative plastic responses that confer either tolerance or avoidance of light competition, thus tailoring their response to prevailing and future competitive circumstances[18, 26]. However, to the best of our knowledge, this hypothesis has not been previously tested.

Here, we examined the ability of plants to shift between these three alternative plastic responses according to their competitive environment. We used an experimental setup with the clonal plant *Potentilla reptans* that simulated, using vertical stripes of transparent green plastic filters, different light-competition settings, by changing both the height and density of the surrounding vegetation. This setup provided a realistic simulation of light competition due to the use vertical filters, which created gradients in both PAR and R:FR particularly in the short treatments. Such use of vertical filters has been shown to induce greater competitive responses compared to homogeneous horizontal shade[23, 40]. Moreover, the use of spaced filter stripes provided plants the possibility to express both vertical and horizontal growth in response to competition cues without any confounding effect of neighbour identity. Here, we predicted that *P. reptans* would exhibit (1) increased vertical inclination (measured as height-per-diameter ratio) when subjected to treatments simulating similarly sized and dense neighbours, which can be outgrown vertically but present limited advantage of horizontal escape; (2) greater shade tolerance responses (measured as specific leaf area) when subjected to treatments simulating taller, and particularly dense neighbours, which offer limited advantages of both vertical or horizontal growth; and (3) increased lateral avoidance (measured as total stolon length and internode length) when subjected to taller but sparse neighbours, which cannot be outgrown vertically but offer greater light availability in the horizontal direction. Our results corroborate these predictions, suggesting that *P. reptans* can shift between three different phenotypes, each of which represents alternative optimal plastic response to a different light-competition scenario.

## Results

**Plant performance**. The performance of *P. reptans* ramets in response to the different treatments, which was measured as the number of newly-produced leaves, was highest under the short-sparse treatment and lowest in the two tall treatments, with intermediate growth under the short-dense treatment (Table 1, Figs. 1a and 2), indicating that plants were negatively affected by simulated competition. In addition, *P. reptans* exhibited greater petiole length and greater height under the dense compared to the two sparse treatments (Table 1, Figs. 1b and 2). The increased height in the dense treatments is also depicted in their shift further along the height–diameter relationship in the standardised major axis (SMA) test (Table 2, Fig. 3a).

**Vertical inclination**. Unlike petiole length and height, the vertical inclination exhibited by *P. reptans*, which was measured as

**Table 1 Generalised linear mixed model results for the effect of filter height and density on *P. reptans* responses**

|  | Height (H) | | Density (D) | | H × D | | Genotype | |
|---|---|---|---|---|---|---|---|---|
|  | F | P | F | P | F | P | Wald Z | P |
| Number of newly produced leaves | **5.407** | **0.023** | 2.030 | 0.159 | 2.333 | 0.131 | 0.661 | 0.508 |
| Petiole length (cm) | 3.563 | 0.063 | **46.774** | **<0.001** | 0.059 | 0.809 | **2.104** | **0.035** |
| Height per diameter | **4.960** | **0.029** | **4.887** | **0.030** | 1.570 | 0.214 | 1.351 | 0.177 |
| Specific leaf area (cm² g⁻¹) | **10.478** | **0.002** | 2.068 | 0.155 | 3.550 | 0.064 | 0.838 | 0.402 |
| Total stolon length (cm) | **4.009** | **0.049** | 0.137 | 0.712 | 4.155 | 0.695 | **2.873** | **0.004** |
| Mean internode length (cm) | 1.797 | 0.185 | 0.269 | 0.606 | 0.344 | 0.560 | **2.207** | **0.027** |

Genotype was used as a random factor (variance estimates presented) and number of leaves as a covariate for total stolon length (F = 3.104, P = 0.082). Significant effects are shown in bold

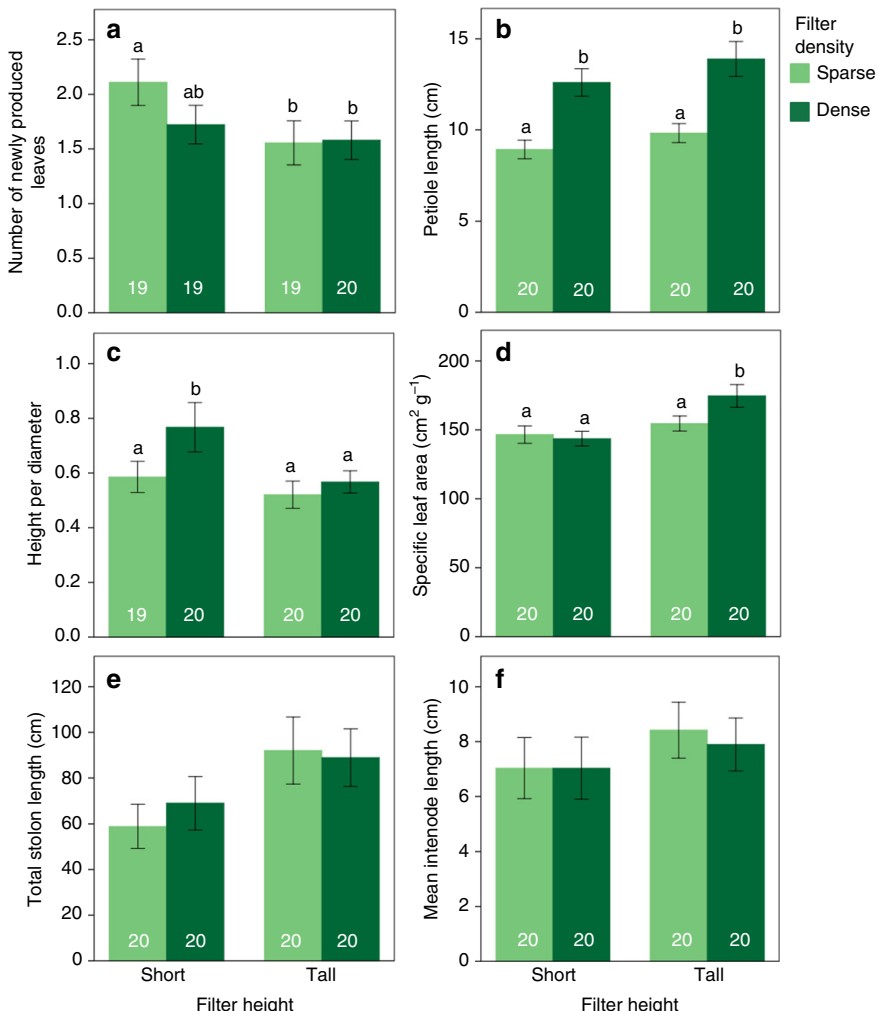

**Fig. 1** Responses of *P. reptans* to the different treatments simulating vegetative shade. Values are mean ± SEM of number of newly produced leaves (**a**), petiole length (**b**), height-per-diameter ratio (**c**), specific leaf area (**d**) total stolon length (**e**) and mean internode length (**f**) in the different treatments (sparse and dense treatments are indicated in light and dark green, respectively). Different letters indicate statistically significant differences between treatments estimated using the least significant difference (LSD) test with the false discovery rate correction[59] following generalised linear-mixed model analyses (see Table 1 for further information). Values of total stolon length are predicted values following a generalised linear-mixed model analysis with number of leaves per plant as a covariate. Sample size per treatment is indicated in white

height-per-diameter ratio, increased not only in response to filter density but also in response to reduced filter height (Table 1). These responses resulted in the greatest vertical inclination when plants were subjected to simulated short-dense neighbours (Figs. 1c and 2; see also Fig. 4a, b for representative photographs of height inclination of plants under the short-sparse vs. short-dense treatments). The greater vertical inclination in response to the short and dense treatments is also depicted in the SMA analysis, where they show greater elevation (y-intercept) and shifts along the slope, respectively (Table 2, Fig. 3a). These results suggest that shifts in plant height in this case were independent of plant diameter. This independence is further confirmed by the

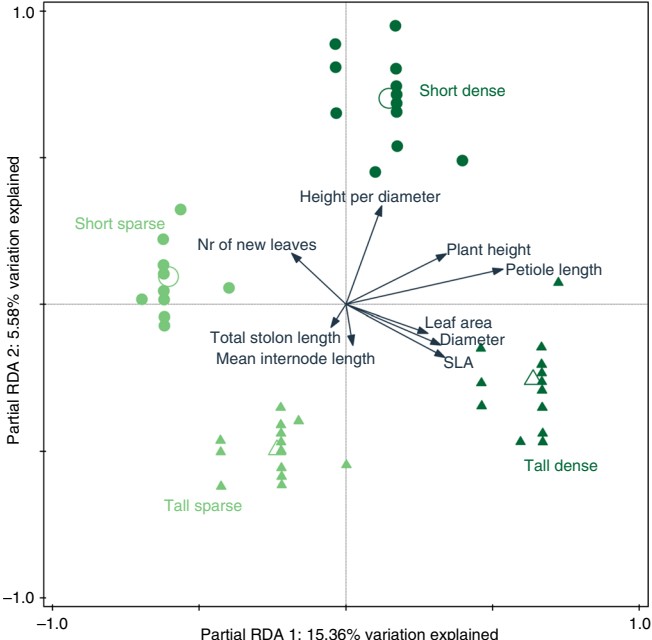

**Fig. 2** Multivariate comparison of the responses of *P. reptans* to the different treatments. Ordination diagram of a partial redundancy analysis, with the explained variation of the first partial RDA axis 15.36% (pseudo-$F = 8.9$, $P = 0.001$, $n = 48$, permutations $n = 999$ for each test) and of the second partial RDA axis 5.58% (pseudo-$F = 3.5$, $P = 0.021$, $n = 48$). Each of the treatments independently accounted for a significant amount of variation in the plastic responses of *P. reptans*: short-sparse (9%, pseudo-$F = 5.1$, $P = 0.002$, $n = 10$), short-dense (5.3%, pseudo-$F = 2.8$, $P = 0.022$, $n = 12$), tall-sparse (4.5%, pseudo-$F = 2.4$, $P = 0.046$, $n = 13$) and tall-dense (11.4%, pseudo-$F = 6.5$, $P = 0.002$, $n = 13$). Arrows represent the assumed linear change of traits as the response variables along the ordination axes. Full symbols represent the individual observations and open symbols the centroids of factor predictors: light green circles, short-sparse; dark green circles, short-dense; light green triangles, tall-sparse; dark green triangles, tall-dense

right angle between the height-per-diameter and diameter response arrows in the ordination diagram (Fig. 2).

**Shade tolerance**. Shade tolerance of *P. reptans*, which was measured as specific leaf area, increased in response to increased filter height (Table 1). Moreover, shade tolerance was particularly high in plants that were subjected to simulated dense and tall neighbours (Figs. 1d and 2). This is also indicated by the nearly significant height × density interaction (generalised linear-mixed model: $P = 0.064$, Table 1) and the clear positive relationship between specific leaf area and the tall-dense treatment in the partial RDA (Fig. 2). These plants also exhibited low vertical inclination and greater plant diameter (Figs. 1c and 2). Specific leaf area was also highly correlated with leaf area (Fig. 2).

**Lateral clonal growth**. *Potentilla reptans* displayed increased lateral clonal growth, with an increase in total stolon length as a response to tall neighbours (Table 1, Figs. 1e and 2), although this trend was not depicted in the SMA analysis (Table 2, Fig. 3b). Mean internode length showed a similar, though non-significant trend of increase under tall and particularly sparse neighbours (Table 1, Figs. 1f and 2). These plants also exhibited low vertical inclination and low shade tolerance (Figs. 1c, d and 2).

**Multivariate analyses**. The partial redundancy analysis revealed that our four simulated neighbour treatments were clearly associated with alternative plastic responses of *P. reptans* (Fig. 2). The variation in the plastic responses of *P. reptans* explained by treatments amounted to almost 22%, with the first two partial RDA axes as well as each of the treatments independently having a significant effect (Fig. 2). Differences in the ramet's size and genotype accounted for 48% of the total variation. Specifically, while plants in the short-sparse treatment were only associated with high performance response (number of newly produced leaves), plants in the short-dense treatment were strongly associated with increased vertical inclination (height-per-diameter); plants in the tall-dense treatment were only associated with increased shade-tolerance responses (leaf area and specific leaf area); and plants in the tall-sparse treatment were associated with neither vertical inclination nor shade tolerance, but showed a trend of increase in horizontal clonal growth (total stolon length and mean internode length, Fig. 2).

## Discussion
Plants can plastically respond to light competition in three well-documented strategies, including 'confrontational' vertical growth, shade tolerance or lateral avoidance, but the alternative ecological contexts that stimulate each of these responses have been seldom studied. Following Novoplansky[18], this study examined the hypothesis that plants are able to 'choose' between these plastic responses according to the relative stature and densities of their opponents. Our results provide support for this hypothesis, demonstrating that *P. reptans* exhibited the highest vertical inclination (i.e. height-per-diameter ratio) when subjected to treatments simulating short-dense neighbours. Conversely, under simulated tall-dense neighbours, ramets of the same genotypes displayed low vertical inclination and the highest shade tolerance via an increase in leaf area and specific leaf area. Finally, under tall and particularly sparse neighbours, these genotypes exhibited both low vertical inclination and low shade tolerance but an increase in total stolon length, suggesting a shift to lateral-avoidance behaviours. *Potentilla reptans* thus displayed a decision-making ability, whereby it could optimally match one of three alternative plastic responses to the prevailing light-competition scenario.

Interestingly, the different phenotypes displayed by *P. reptans* in our experiment clearly corresponded to differences in PAR and R:FR levels between treatments, thus providing an insight as to the information used by plants to assess their competitive environment and the optimal strategy they should tailor to it. Specifically, the strong shade-tolerance response displayed by *P. reptans* under the tall-dense treatment coincides with the homogenously low PAR and R:FR levels created by this treatment (Fig. 2). In contrast, the vertical inclination displayed mainly under the short-dense treatment can be attributed to a gradient of increased R:FR levels, which was mainly experienced by plants within this treatment (Fig. 2). A few previous studies have demonstrated that plants can indeed perceive- and readily respond to gradients of resource availability compared to prevailing levels of either light[23, 40] or nutrients[41, 42]. For example, Weijschedé et al.[23] showed that *Trifolium repens* exhibit greater petiole elongation under a vertical light gradient compared to homogeneously intense shade, and that such elongation was also more beneficial under the light gradient. Interestingly, Vermeulen et al.[36] have shown that when *P. reptans* is presented with a gradient of light competition, petiole length in some genotypes can reach up to 50 cm, which is the height of our simulated tall neighbours. This result supports the notion that in *P. reptans*, vertical inclination of the petioles is indeed more responsive to vertical light gradients compared to homogeneous shade.

**Table 2 Standardised major axis (SMA) results for the effect of treatments on the allometric relationships between plant traits**

| | Plant height vs. diameter | | | | Total stolon length vs. number of leaves | | | |
| | Shift along a common slope | | Difference in elevation | | Shift along a common slope | | Difference in elevation | |
| Effect | Wald | P | Wald | P | Wald | P | Wald | P |
|---|---|---|---|---|---|---|---|---|
| Height | 2.340 | 0.126 | **3.878** | **0.049** | 0.008 | 0.928 | 2.420 | 0.108 |
| Density | **11.430** | **<0.001** | 0.268 | 0.605 | 0.196 | 0.658 | 0.299 | 0.584 |

Included are tests for shifts along a common slope and differences in elevation (significant effects are shown in bold)

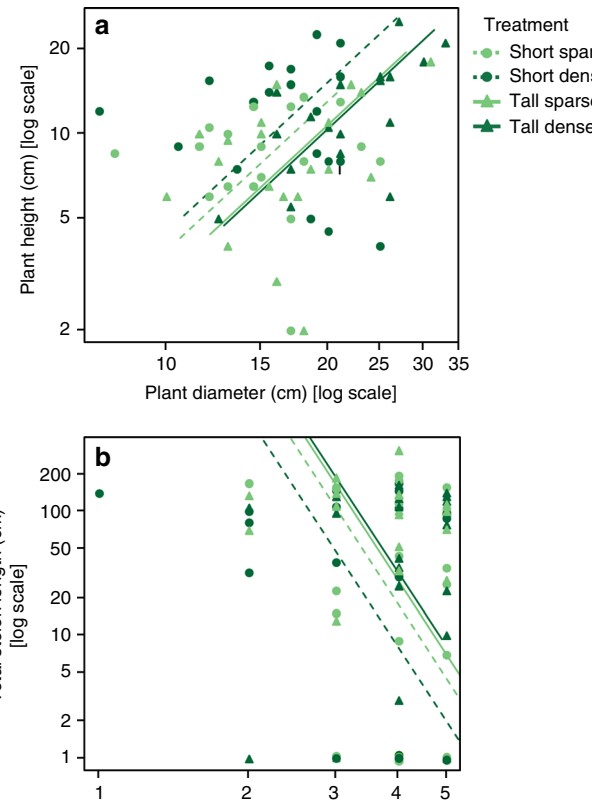

**Fig. 3** Treatment effects on the allometric relationships between plant traits. Standardised major axis (SMA) relationships between plant height and diameter (**a**, $n = 19$) and between total stolon length and number of leaves (**b**, $n = 20$) within the short (circles and dashed lines), tall (triangles and solid lines) sparse (bright green) and dense (dark green) treatments (see Table 2 for further information)

Although the sensory mechanisms with which plants can assess such resource gradients are still unknown, the selective advantage of this perception is clear, as it could provide information of future resource levels and competition[43, 44].

Unlike their vertical inclination, petiole length of *P. reptans* increased under both dense treatments, regardless of their height. This result might imply that increased petiole length in *P. reptans* is an early response to crowding, with which plants can assess potential increments in light intensity. Following such initial elongation, plants under the tall-dense treatment might have refrained from further elongation or vertical inclination, which could entail construction and maintenance costs[29, 45]. Instead, these plants invested in a shade-tolerance strategy via increased specific leaf area, which requires low construction costs. Further

studies are therefore required to learn whether in *P. reptans* or other species, increased petiole length is indeed an initial response to dense stands.

In partial support of our prediction, *P. reptans* exhibited a trend towards lateral avoidance via clonal growth under competition with simulated tall, and particularly sparse, neighbours, i.e. in a scenario where horizontal escape from competition was possible. In addition, these plants did not show either increased vertical inclination or shade tolerance responses, indicating that this third phenotype tended to laterally avoid neighbours that were too tall to overtop but not too dense to not allow lateral escape. The fact that this plastic response in clonal growth was somewhat weaker than the prominent shifts exhibited by *P. reptans* in vertical inclination and shade tolerance, could be attributed to the high fitness costs associated with maintaining the physiological machinery required for such plasticity[46]. This limited plasticity might also reflect a reduced selective advantage of responsiveness to horizontal light availability. Unlike the aforementioned predictable pattern of vertical light gradients, variability in light availability in the horizontal direction might be less predictable[47–49], especially in plants from spatially and temporally heterogeneous environments. This notion can be supported by results of previous studies that have shown reduced plasticity of stoloniferous plants in horizontal compared to vertical spacers in response to light availability[47, 50]. Such limited responsiveness to variations in horizontal light availability might also explain the lack of differences found in our experiment in total stolon length between the sparse and dense competition treatments. It is thus possible that clonal plants such as *P. reptans* are more responsive to heterogeneity rather than mere availability of light at the horizontal direction, as the former might provide a more reliable cue for the probability of finding unoccupied light patches.

In conclusion, the results of this study offer a contextual framework for the different well-known responses of plants to light competition. More importantly, we have demonstrated a decision-making ability in plants, which allows them to adaptively 'choose' between three responses, according to the relative stature and density of their opponents. This ability of plants to exhibit plasticity in their plastic responses, or 'metaplasticity' (*sensu* Novoplansky[18]), might not only be restricted to competition-related behaviours but also encompass decision-making in plants in response to other factors, such as resource availability. For example, Cahill et al.[51] showed that root foraging decisions of *Abutilon theophrasti* in response to resource heterogeneity is contingent upon the presence of neighbours. Both the results of Cahill et al.[51] and those of our study suggest that plants are capable of acquiring and integrating complex information about their environment in order to adaptively modify their extent of plastic responses. Such complex decision-making in plants could have important implications on our understanding of the processes that govern plant behaviour.

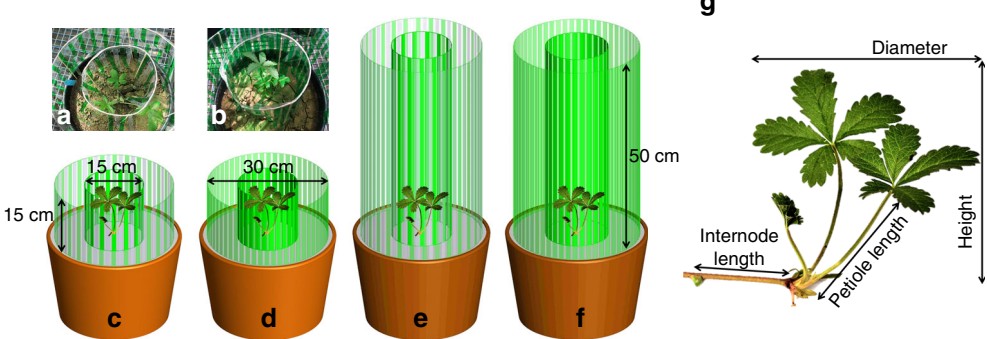

**Fig. 4** Graphic representation of the experimental design and measured variables. The experiment included treatments that simulated short-sparse **a,c**, short-dense **b,d**, tall-sparse **e** and tall-dense **f** neighbouring vegetation. Included are representative photographs of the short-sparse **a** and short-dense **b** treatments with *P. reptans* in the middle of the pots. Dense vegetation **d,f** was simulated using strips of transparent green plastic filters that mimic vegetative shade in both light transmission levels and R:FR ratios, while sparse vegetation **c,e** was simulated with alternating green and clear strips (see Materials and Methods). A photograph of *P. reptans* is included with a description of some of the measured variables **g**

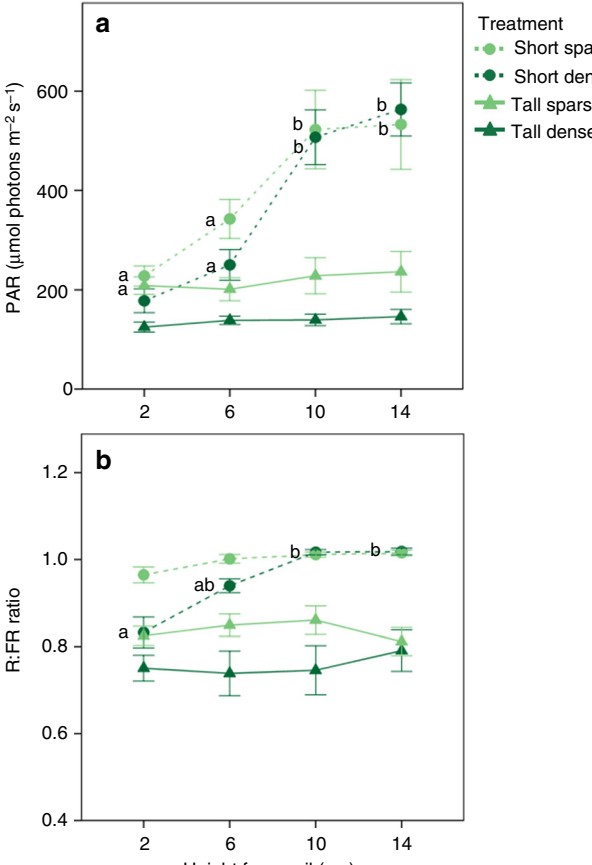

**Fig. 5** Light characteristics in the different treatments simulating vegetative shade. Values are means ± SEM of photosynthetically active radiation (PAR; **a**) and red to far-red ratio (R:FR; **b**) at different heights within the short (circles and dashed lines), tall (triangles and solid lines) sparse (bright green) and dense (dark green) treatments. Different letters indicate statistically significant differences between heights within a treatment, estimated using the least significant difference (LSD) test with the false discovery rate correction[59] following an ANOVA in **a** (ANOVA, height from soil: $P < 0.001$, $n = 7$) and **b** (ANOVA, height from soil: $P = 0.006$, $n = 8$)

## Methods

**Plant material**. *Potentilla reptans* L. is a stoloniferous perennial herb, which occurs in meadows and river banks as well as disturbed environments, such as roadsides

and pastures[52]. It grows stolons with rooted ramets, which normally stay connected to the mother plant for one season[52] and has been shown to have high levels of clonal integration[53]. The ramets are rosettes that form indeterminate number of leaves whose height is regulated by petiole length and vertical inclination. Leaf height has been shown to increase with that of neighbouring plants[36, 47], while leaf number has been shown to decrease under shade[54, 55].

Twenty *P. reptans* cuttings from different sites around Tübingen, Germany were collected in December 2013. To relax environmentally-induced maternal effects, the cuttings were clonally propagated for one generation under common-garden conditions at a field site in Tübingen University, where they were individually planted in 1 L pots and covered with light fabric organza during winter.

**Experimental setup**. In May 2014, four newly grown ramets were severed from each *P. reptans* mother plant. Each ramet was then planted separately in the centre of a 12 L pot (30 cm diameter) filled with local topsoil (Bischoff GmbH & Co. KG, Rottenburg am Neckar, Germany). Individual ramets, one ramet per genotype, were assigned to one of four light-competition treatments simulating different height and density of neighbours. Light competition was simulated using transparent green plastic filters, which mimic vegetative shade in both light transmission levels and R:FR ratios (122 Fern green, Lee filters, CA, USA)[23, 36]. One centimetre-wide filter strips were used to create a grid of two concentric cylinders of 15 and 30 cm in diameter (Fig. 4a–f). The use of two cylinders rather than a single one provided a better simulation of light competition that could be experienced by clonal plants as they expand horizontally. To simulate vegetation through which plants could grow laterally, the filter strips were positioned 0.5 cm apart. Short and tall neighbours were simulated using 15- and 50-cm-long filter strips, respectively (Fig. 4a–f). Sparse neighbours were simulated with alternating green and clear (130 clear, Lee filters, CA, USA) strips (Fig. 4a, c, e), while dense neighbours were simulated using green strips only (Fig. 4b, d, f). This experiment resulted in a total of 80 plants (20 genotypes × 2 density treatments × 2 height treatments). The sample size ($n = 20$) was chosen based on previous studies where developmental plasticity in *P. reptans* in response to light competition was shown for a smaller sample of ca. 10 genotypes[36, 55].

The filters were set around the plants following transplantation and the pots were placed on benches at a greenhouse in Tübingen University, within a distance of 50 cm between pots, to prevent shading effects among neighbouring plants. The plants were arranged in blocks according to genotype and assigned random numbers to conceal their identity during variable measurements. Photosynthetically active radiation (PAR) was measured on September 2015 for a subset of pots (seven per treatment) at four heights from the soil (2, 6, 10 and 14 cm) using a quantum light metre (LI-250, LI-COR Inc., Lincoln, NE, USA). PAR differed among treatments (ANOVA: $F_{3,96} = 37.087$, $P < 0.001$), height from soil (ANOVA: $F_{3,96} = 18.167$, $P < 0.001$) and their interaction (ANOVA: $F_{9,96} = 4.883$, $P < 0.001$). At a height of 2–6 cm, which is leaf-lamina height at the beginning of the experiment, PAR was higher in the short-sparse compared to the tall-dense treatment, while the short-dense and tall-sparse treatments had similar intermediate PAR levels (Fig. 5a). However, the two short treatments were characterised by a gradient of increased PAR with increasing height, while PAR levels at the two tall treatments remained low (Fig. 5a). In addition to PAR levels, red to far-red ratio (R:FR) was measured in July 2017 for a subset of pots (eight per treatment) at the same four heights using a FieldSpec 4 Standard-Res Spectroradiometer (ASD Inc., Longmont, CO, USA). R:FR differed among treatments (ANOVA: $F_{3,112} = 53.949$, $P < 0.001$), height from soil (ANOVA: $F_{3,112} = 4.327$, $P = 0.006$) and their interaction (ANOVA: $F_{9,112} = 2.018$, $P = 0.044$). As for PAR levels, R:FR was highest in the short-sparse treatment and lowest in the tall-

dense treatment (Fig. 5b). Here however, only the short-dense treatment showed a gradient of increased R:FR with increasing height (Fig. 5b). Hence, in summary, each of the treatments exhibited a unique light environment, which was expressed in light intensity, light quality or both, and/or a gradient in these light characteristics.

**Measured variables**. Ramet performance in response to the different treatments was estimated from the number of newly produced leaves per ramet, i.e. number of leaves at the end of the experiment minus number of leaves at its onset. Leaf number has been shown to provide an appropriate non-destructive estimate of plant size in *P. reptans*[54], and was also found to highly correlate with ramet biomass in a separate experiment (Supplementary Fig. 1a). The numbers of newly produced leaves in one ramet in the short-sparse, short-dense and tall-dense treatments were lost due to technical problems, resulting in a sample size of 19 per treatment.

Measurements of the plastic responses of *P. reptans* were carried out in August 2014, 9 weeks after the onset of the experiment. These measurements included petiole length as well as plant height, which was estimated as the vertical distance between the highest leaf tip and the soil surface. Vertical inclination in response to competition was evaluated as the ratio between plant height and its diameter, i.e. the maximum distance between the two furthermost leaf tips. This ratio was chosen because it proved easier to measure within the shading apparatus compared to petioles angles, and due to its high correlation with the latter (Supplementary Fig. 1b). Plant height in one ramet in the short-sparse treatment was lost due to technical problems, resulting in a sample size of 19 for this treatment.

Shade tolerance was estimated with specific leaf area, i.e. the ratio between lamina area and its dry weight. To that end, the two biggest laminas per ramet were harvested and photographed, and their images were used to quantify mean lamina area with the ImageJ software[56]. Laminae's biomass was measured following oven drying them in 70 °C for 3 days. Lateral clonal growth was estimated by measuring total stolon length per plant as well as mean internode length of the stolons.

All variables were measured in the pots prior to the removal of the filters so as to not disrupt plant architecture, except for lamina area and biomass. For the latter variables, samples were identified according to randomly assigned numbers rather than treatment names to blind the investigator to treatment identity. All collected data are available in the Supplementary Data.

**Data analysis**. The responses of *P. reptans* to the height and density of simulated neighbours expressed by their number of newly produced leaves, petiole length, height-per-diameter ratio, specific leaf area, total stolon length and mean internode length were examined using a generalised linear-mixed model with filter height and density as fixed factors and plant genotype as a random factor. The number of newly produced leaves, petiole length, height-per-diameter ratio and specific leaf area were analysed with a normal probability distribution with an identity link function. Total stolon length and internode length were analysed with a Gamma probability distribution with a log link function. To account for potential differences in total stolon length due to ramet size[57, 58], the number of leaves per ramet at the end of the experiment was used as a covariate (after confirming the assumption of homogeneity of slopes). Post hoc pairwise comparisons between treatments were performed using false discovery rate correction for multiple tests[59, 60]. These statistical analyses were performed using PASW 18 (SPSS).

In addition to the univariate analyses, a multivariate approach was employed to evaluate the complete array of plastic responses displayed by *P. reptans* under the different treatments. Partial redundancy analysis (RDA) was performed with diameter, height per diameter, leaf area, mean internode length, number of new leaves, petiole length, plant height, specific leaf area and total stolon length as response variables explained by the simulated neighbour treatments after removing the effect of covariates (number of leaves at the end of the experiment as an estimate of plant size and genotype). To account for different measuring units, traits were centred and standardised. Monte-Carlo permutation test ($n = 999$) on first and second RDA axes was used with genotype as permutation block[61]. Furthermore, to evaluate the effect of each of the treatments independently, simple tests for each treatment were performed using the false discovery rate correction for multiple tests[50, 62]. The multivariate analyses were performed using CANOCO 5[63].

To better estimate the treatment effects on the allometric relationships between plant height and diameter as well as between total stolon length and number of leaves, these relationships and effects were analysed using the SMA regression, which is appropriate for analysing variables that have no causal relationship[64]. SMA was used to test for the effect of filter height and density on shifts along a common slope and on elevation (*y*-intercept) between slopes (after confirming the assumption of homogeneity of slopes). The traits were log-transformed prior to the analysis. These analyses were performed using the smart package in R[62].

**Data availability**. Data analysed in this study are included in the Supplementary Information

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

## Acknowledgements
We thank Nadine Bihler, Elena Spöri, Sarah Gebauer, Ricarda Gatter, Peter Armstrong and Lorenz Henneberg for their assistance with the experimental setup; Pierre Liancourt, Jan Ruppert and Udi Segev for valuable discussions and contributions to this work; and Shimon Gruntman for producing the graphic illustrations of the experiment. This study was supported by a grant of the Tübingen Athene Program to M.G.

## Author contributions
Design of the research (M.G. and D.G.); performance of the research (D.G.); collection and data analysis (D.G., M.M. and M.G.); writing the manuscript (M.G., D.G., M.M. and K.T.).

## Additional information

**Competing interests:** The authors declare that they have no competing financial interests.

