## [Peer Review File · Nature Communications]

Reviewers' comments:

Reviewer #1 (Remarks to the Author):

This Paper uses an interesting experimental design that simulates shade of the leaves of neighbouring plants using green filters. The filters are either sparse or dense, and short or tall; simulating different types of competition from neighbours that might be either sparse or dense, and short or tall. It ties together a number of well understood physiological responses with ecological results and has the potential to be an important contribution to the literature. I found the paper quite interesting, easy to read, and well organised. I have a couple of questions about the GLMMs and the presentation of the ordination that I'd like the authors to consider, but these are relatively minor.

However, I do find the framing of the paper as if all of these responses are new and unexpected to be a bit unhelpful, and I feel like it detracts from the importance of the results. The study as I understand the literature, actually ties together a number of well understood threads of theoretical, ecological and physiological understanding. But rather than write the paper as if the authors were achieving this important synthesis and test of theory, the authors claim that everything they found was novel. I know that is the trend in how we are all expected to write our papers, but in this case I feel like it detracts from the true novelty which is tying together all these loose threads to generate results that are highly predictable. I think that most of these results are predictable in a qualitative sense: R/FR causes elongation, PAR levels change SLA, and neighbouring leaves can influence both R/FR and PAR levels. I want to be very clear, this is not meant to be a criticism, to me the fact that our understanding of plants is so sophisticated that we can make these sorts of predictions and syntheses is extremely exciting! Scientists predictions based on existing knowledge, and experimental tests of our predictions that fail to be rejected should be exciting! We shouldn't feel the need to write our papers as if we were surprised and everything is new. It seems to me that this paper is very appropriate for Nature Communications. But it would benefit greatly from framing the problem as a synthesis of many diverse threads of understanding, producing an important vertical increase in knowledge of the field, instead of being framed as completely novel findings that arose de novo from the greenhouse. I want to emphasize that this isn't a criticism, it's an invitation to strengthen the paper to have the maximum impact on our field and the literature as a whole. I look forward to seeing this paper in press.

ABSTRACT:

Line 7: I think in some sense every experimental setup is novel. We're generally in the business of pushing the boundaries of knowledge, and so – by definition- nearly everything scientists do is novel. Folks have been using red light filters to simulate competition and study the R/FR response of plants since the 1980s. Whether or not the filters were arranged in this particular fashion before, hardly seems like the selling point of these highly interesting results.

Lines 11-13: I'm not sure this is the first study to show this. Several that are cited by the authors show complex plant responses to all sorts of competitive environments and a diversity of cues. Mora Gersani has many interesting papers on this subject, many with Ariel, and perhaps some of the authors as a co-author. Again, this is not meant as a criticism, the results are in my opinion important and exciting. What difference does it make if they are or, are not "first"? We're contributing to a vertical increase in knowledge by building on what we already know.

MAIN TEXT

I found the introductory text to be quite easy to read, interesting, and that it does a good job of setting up the experiment. It could potentially benefit from a slightly more in-depth review of what is known about: 1) the well understood R/FR responses that plants have via phytochrome b, and; 2) the well understood shade vs sun leaf responses that many plants show. The paper would be

strengthened in my opinion by reviewing this understanding and using it to frame hypotheses which I imagine were behind the very clever experimental design.

Line 81: There's that word novel again.

RESULTS

The results section is clear and easy to understand, I don't have too many specific comments. The exciting part of this paper is the clean and interesting results that show plasticity of plant growth, and the sophisticated ways that plants are able to assess and then respond to cues associated with competition.

Theory predicts a decrease in the ratio of red to far-red light would cause plants to increase in height. The authors saw this. It also predicts that plants should produce shade leaves with higher SLA under low light. The authors saw this. I'm not sure anyone would be surprised at these responses at a qualitative level – which I stress is not a criticism. What I find exciting is that their clever experimental design with sparse/dense and short/tall light filters allows both of these results to be placed along a gradient of simulated competition in an ecological context which has the potential to greatly improve our understanding of the ecological impacts of these physiological responses. How cool! I wish the paper had been framed this way, rather than as if all of this were completely "novel".

Figs: Is all green the best choice considering some readers might be colour blind? Would these appear as light and dark grey to a colour blind person? Maybe that's ok.

Fig 3: I think the centroids for sparse, short, dense and tall are probably more confusing than helpful. I think having the four treatment centroids is probably more useful to readers. I don't feel too strongly about this though.

Fig 3: However, I would like to see the individual data points on here not just the centroids; that seems important. Perhaps with ellipses around each treatment point cloud. This seems very important to me.

Fig 3: Also, 22% of variance explained isn't a lot. I don't think this sinks the paper, but it would be helpful to include some discussion of what else might be going on for the other 80% of the variance in the data.

DISCUSSION:

The discussion is quite long. Consider a bit more review of plant responses to R/fr and PAR in the introduction instead of such a long discussion.

Line 156-158: It seems to me that the information plants use to adjust internode lengths, height, and SRL in response to R/FR and light intensity are almost completely understood, and have been since the 1980s or even earlier. Again, this study shows the complexity of plant responses, and casts all of these exceptionally well understood responses in an ecological competition light, which is cool. I just don't understand why the paper is written as if everything is novel. Your data are exciting! It seems to me you could just let them stand on their own and our field would be better served by your important contribution.

METHODS:

Line 235: I assume this plant is perennial? Please state this somewhere. Short studies of very long lived perennials are perfectly valuable, but these data only show short term responses during the establishment of very young ramets. Readers should know that the lifetime responses of long lived

plants might be very different if this plant were allowed to live out its life (which could be decades or centuries long).

Line 310: Given that PAR varied 5-fold between treatments; I'd be curious to see the GLMMs with PAR as a co-variate. Detangling the effects of R/FR from PAR intensity would greatly strengthen the interpretation of results. The expectation would be that low PAR produces shade leaves, and that high R/FR produces avoidance responses. But perhaps they interact! This would be exciting!

Line 311: If the experiment was organised as a random block design, why isn't block a random effect in the GLMMs?

Reviewer #2 (Remarks to the Author):

Review Nature Communications 2017

This manuscript makes the exciting claim that plants show varied, sophisticated adaptive responses to different competitive environments, responding to light cues. It reports responses of *Potentilla reptans* to novel artificial shading treatments. The manuscript argues that 1) the shading setup produces complex light cues that mimic different competitor heights and densities found in nature, 2) *Potentilla reptans* produces three very different responses to competitors in response to these treatments, and 3) these responses are likely adaptive to natural competition. If some plants can tailor their responses to competitors to laterally avoid shading, tolerate shade, or vertically avoid shading depending on competitive environment, then plants can show sophisticated behaviour comparable to that of animals.

Evaluation of these three arguments.

1. Argument 1) I agree that the experimental setup, by using a shade filter that lowers R:FR and creating strips that mimic plant stems, is appropriate methodology for mimicking different types of competition. The authors should report how R:FR differs among treatments, as well as irradiance, since low R:FR is the predominant light cue of competitors.

2. Argument 1) Note that while the setup in using strips of filter and using 2 cylinders of strips provides a more realistic simulation of light competition than would the same combinations of irradiance and R:FR created by overhead filter sheets (the most common way to simulate competition), they have not provided experimental evidence or references that indicate the realism makes a difference.

3. Argument 3) I agree that the adaptive interpretation of the three behaviours is consistent with the current consensus given functional arguments, comparisons of how population and species from different competitive environments respond to low R:FR and irradiance, and measures of natural selection for vertical growth.

4. Argument 3) The authors could be better explain the traits and their function to an audience unfamiliar with these traits - see below.

5. Argument 2) I have concerns with the data interpretation related to the claim of responses specific to each type of competition. For petiole length and stolon length, the interpretations do match with the statistics in Table 1 and means in Fig 2. However, for height-per-diameter and specific leaf area, the statistics do not support the interpretation. The wording of the manuscript implies the evidence for three different competitive responses is that the responses depend on the combination of filter density and height. Such a specific response would be indicated by a significant Height X Density interaction in the analyses of variance. While the means for height-per-diameter and specific leaf area reported in in Figure 2 follow the patterns expected from Height X Density interactions, the statistical analysis shows that the Height X Density interactions are in fact not significant.

a. "Unlike plant petiole length and height, the vertical inclination exhibited by *P. reptans*, which was measured as height-per-diameter ratio, was greatest when plants were subjected to simulated short-dense neighbours (Table 1, Fig. 2c, Fig. 3; ". Here, Table 1 indicates significant responses in height per diameter to Height and Density, but no significant HeightXDensity interaction. Fig 2c

indicates that height per diameter is greatest in short dense, and differs significantly from tall sparse and tall dense, but not short sparse.

b. "In contrast to their vertical inclination, *P. reptans* exhibited the highest shade tolerance, which was measured as leaf area and specific leaf area, when subjected to simulated dense and tall neighbours (Table 1, Fig. 2d, Fig. 3)" Here Table 1 indicates a significant effect of Height and a nearly significant Height X Density interaction for specific leaf area. Figure 2d indicates that the specific leaf area for tall dense differs significantly from that of short sparse and dense, but not from tall sparse. Figure 3 indicates that specific leaf area and leaf area responses are highly correlated.

6. Argument 2) This mismatch between how the treatments appear to differ and whether the statistics back up that appearance needs to be resolved such that the interpretation follows the statistical analysis. Here are possible approaches, with the first two related to further statistics, and the second two at changing the interpretation of the present statistics: 1) It would be useful, if not already done, to check that the residuals meet the assumptions of anova, particularly for homoscedasticity. 2) The partial redundancy analysis is interesting, but purely descriptive. However, the authors could look for the interactions in multivariate space. 3) The interpretation could be based on the significant main effects only. 4) The interpretation could note that the main effect only is significant, and that the main effect is clearly resulting from the larger mean in a particular treatment combination.

Detailed comments:

1. The paper structure can be tightened up and made more concise, less review-style. In particular, the first paragraph of paper is short and weak. Foreshadow the direction of the paper - that there are different kinds of responses to competitor - tolerate, evade, avoid, resist, escape . The question is how precisely can plants tailor their response to the kind of competitor present. Is it simply genetic differentiation in which traits are cued by low R:FR, or can plants sense more than low light and low R:FR indicating a mass of chlorophyll and respond adaptively to complex cues from light
2. Terminology: Rather than calling it a shift, call it phenotypic plasticity, developmental plasticity, response to the environment. Be specific that the experiment examines looking at responses to light cues.
3. Terminology: Petiole length is not synonymous with petiole elongation. While greater petiole elongation causes longer petioles, but larger leaves and older leaves will have longer petioles. To be a measure of elongation, the length must be relativized to either lamina size (mass or area), aboveground mass, petiole mass, or other measure of size. For the current trait, use petiole length throughout.
4. Terminology: Authors need to be consistent in discussing vertical growth - the same trait is variously identified as vertical inclination and height per diameter. Petiole length is sometimes included as a component of vertical growth, but sometimes not. Vertical inclination itself needs defining as a term.
5. Missing reference: Crepy, M. A., and J. J. Casal. 2014. Photoreceptor-mediated kin recognition in plants. *New Phytologist*: on complex light cues that indicate identity.
6. Missing reference: Huber, H., A. Fijan, and H. J. During. 1998. A Comparative Study of Spacer Plasticity in Erect and Stoloniferous Herbs. *Oikos* 81:576-586 on vertical vs. horizontal spacers,
7. A plant diagram is needed that identifies the traits measured. Most people are not familiar with rosette plant traits.
8. The introduction should also focus on the traits measured. For example, the paragraph on vertical growth focuses on stem elongation and shade avoidance. While petiole elongation is measured, there is no discussion of vertical inclination/petiole angle.
9. Figures 1,3,and 4 are difficult to read in black and white. Many symbols and lines are nearly or completely invisible.

Reviewer #3 (Remarks to the Author):

This paper claims to show that plants show distinct strategies in response to vegetation height: shade tolerance in tall vegetation and competitive shade avoidance in short vegetation. Proxies of these two strategies are increases in SLA and in length/diameter, respectively. My main objection to this set up is that increases in SLA are not really a shade tolerance strategy. Across species shade tolerant species typically exhibit lower SLA and less shade-induced plasticity than light demanding species. This makes sense as high SLA is associated with less durable leaves and relatively greater chances of leaf loss through damage that is detrimental in the shade. Furthermore I find the result for the shift in the relation between petiole length and diameter with a $P = 0.049$ rather weak. To confirm whether such low significance is really consistent I would want to see a second experiment.

Reviewer 1

“I do find the framing of the paper as if all of these responses are new and unexpected to be a bit unhelpful, and I feel like it detracts from the importance of the results. The study as I understand the literature, actually ties together a number of well understood threads of theoretical, ecological and physiological understanding. But rather than write the paper as if the authors were achieving this important synthesis and test of theory, the authors claim that everything they found was novel. I know that is the trend in how we are all expected to write our papers, but in this case I feel like it detracts from the true novelty which is tying together all these loose threads to generate results that are highly predictable. I think that most of these results are predictable in a qualitative sense: R/FR causes elongation, PAR levels change SLA, and neighbouring leaves can influence both R/FR and PAR levels. I want to be very clear, this is not meant to be a criticism, to me the fact that our understanding of plants is so sophisticated that we can make these sorts of predictions and syntheses is extremely exciting! Scientists predictions based on existing knowledge, and experimental tests of our predictions that fail to be rejected should be exciting! We shouldn't feel the need to write our papers as if we were surprised and everything is new. It seems to me that this paper is very appropriate for Nature Communications. But it would benefit greatly from framing the problem as a synthesis of many diverse threads of understanding, producing an important vertical increase in knowledge of the field, instead of being framed as completely novel findings that arose de novo from the greenhouse. I want to emphasize that this isn't a criticism, it's an invitation to strengthen the paper to have the maximum impact on our field and the literature as a whole. I look forward to seeing this paper in press.”

We agree with the reviewer that the results of this study are significant in the synthesis they provide regarding the competitive environments that induce the different plastic responses, rather than in the plastic responses themselves. We also apologize if the excessive use of the word ‘novel’ may have oversold our results. In accordance with the reviewers’ suggestion we removed words such as “novel” and “unexplored” from the text and stressed the fact that the traits we studied are already well known (see e.g., L21, 37, 159), and that this study offers a contextual framework for these responses (L159-160, 232-233). While we hope these changes strengthened the focus of the manuscript, as suggested by the reviewer, we still believe that the main importance of our results lies more in their demonstration of decision making abilities in plants, i.e., the fact that plants can decide between and not only within these responses is new, and may open new avenues for ecological, evolutionary and physiological research. We therefore chose to keep this aspect as the main motivation and novelty of our study. We have thus carefully revised the entire manuscript to clearly distinguish between the aspects that are known but have not been synthesized conceptually, and the new finding that a plant can decide between three different responses based on the evaluation of the competitive abilities of its neighbors. We hope that this distinction also helps to better highlight what we believe is the most important finding of our study and to distinguish between our new finding and the aspects that have been studied before and for which we present a synthesizing framework.

“Line 7: I think in some sense every experimental setup is novel. We're generally in the business of pushing the boundaries of knowledge, and so – by definition- nearly everything scientists do is novel. Folks have been using red light filters to simulate competition and study the R/FR response of plants since the 1980s. Whether or not the filters were arranged

in this particular fashion before, hardly seems like the selling point of these highly interesting results.”

According to this comment we removed the word “novel” from the text.

“Lines 11-13: I’m not sure this is the first study to show this. Several that are cited by the authors show complex plant responses to all sorts of competitive environments and a diversity of cues. Mora Gersani has many interesting papers on this subject, many with Ariel, and perhaps some of the authors as a co-author. Again, this is not meant as a criticism, the results are in my opinion important and exciting. What difference does it make if they are or, are not “first”? We’re contributing to a vertical increase in knowledge by building on what we already know.”

We acknowledge again and agree, that our previous version may have suffered from an excessive use of the word “novel”. As explained above, we more clearly highlighted that the most important finding of our experiment was the fact that plants can decide between three different responses to competition and that these are likely to be adaptive. We therefore still think that our results regarding this ability of plants to choose between different plastic responses in accordance with the stature of their neighbors has not been shown previously. While previous studied, such as the ones suggested by the reviewer, have shown that plants can choose wheatear or not to engage in competition according to their genetic relatedness to their neighbors, they did not study the ability of plants to shift between different strategies. Therefore, we have retained the wording in this place. While we agree that sometimes overselling findings can be tiresome, we do believe that highlighting novelties can also be helpful. It can more clearly demonstrate new avenues for future research, e.g. in our case the study of the physiological mechanisms by which plants can switch between three very different growth responses.

“The introductory text could potentially benefit from a slightly more in-depth review of what is known about: 1) the well understood R/FR responses that plants have via phytochrome b, and; 2) the well understood shade vs sun leaf responses that many plants show. The paper would be strengthened in my opinion by reviewing this understanding and using it to frame hypotheses which I imagine were behind the very clever experimental design.”

According to this comment we provided a more thorough review of known shade avoidance and shade tolerance responses in plants and the cues that induce them (L18-46).

“Line 81: There’s that word novel again.”

According to this comment we removed the word “novel” from the text.

“RESULTS: Theory predicts a decrease in the ratio of red to far-red light would cause plants to increase in height. The authors saw this. It also predicts that plants should produce shade leaves with higher SLA under low light. The authors saw this. I’m not sure anyone would be surprised at these responses at a qualitative level – which I stress is not a criticism. What I find exciting is that their clever experimental design with sparse/dense and short/tall light filters allows both of these results to be placed along a gradient of simulated competition in an ecological context which has the potential to greatly improve our understanding of the ecological impacts of these physiological responses. How cool! I wish the paper had be framed this way, rather than as if all of this were completely “novel”.”

We agree with the reviewer and again attempted to more clearly distinguish between the interesting finding regarding decision-making and the single physiological or morphological responses. In the results we aimed to synthesize and emphasize that the alternative plastic responses in *P. reptans* were tied to the alternative cues presented by the different competitive scenarios, i.e. we strengthened the link between plant traits and ecology. We have now tried to frame the discussion to comply with this suggestion (e.g. in L158-172).

“Figs: Is all green the best choice considering some readers might be colour blind? Would these appear as light and dark grey to a colour blind person? Maybe that’s ok.”

In accordance with the suggestions from reviewer 1 and 2, we now use colorblind safe and print friendly colors. We also verified the choice by a color blind colleague and by printing the figures in black and white. We decided to retain ‘green’ as the main color because it best reflects our experiment (i.e. the filters).

“Fig 3: I think the centroids for sparse, short, dense and tall are probably more confusing than helpful. I think having the four treatment centroids is probably more useful to readers. I don’t feel too strongly about this though.”

“Fig 3: However, I would like to see the individual data points on here not just the centroids; that seems important. Perhaps with ellipses around each treatment point cloud. This seems very important to me.”

In accordance with these suggestions, we now use the individual data points together with the centroids of the four treatments. Indeed, both suggestions improved the informative value of the ordination diagram. Because the individual data clouds are well defined among the treatments, we refrained from further adding ellipses around them. Rather, we kept the color and symbol codes consistent among all presented figures to facilitate the understanding of the presented results.

“Fig 3: Also, 22% of variance explained isn’t a lot. I don’t think this sinks the paper, but it would be helpful to include some discussion of what else might be going on for the other 80% of the variance in the data.”

We thank the reviewer for the concern. The reported variation explained by treatments is the variation after removing the effect of covariates. We knew a priori that the plant size and different genotype will contribute to a considerable amount of variation and could potentially confound the effect of the treatments themselves. Indeed, from the 78% of the leftover variation, the differences in plant size and genotype account for 48% (L148-149). This leaves 30% unexplained. One of the possible explanations is that, even though the traits we use are good proxies of complex strategies and trade-offs involved in the different plastic responses to the competition for light, they are still just proxies, i.e. simplified and measurable characters of a more complex physiological and morphological changes.

“The discussion is quite long. Consider a bit more review of plant responses to R/fr and PAR in the introduction instead of such a long discussion.”

In accordance with this suggestion, we shortened the discussion by removing one paragraph, which we believed was superfluous. In line with a previous suggestion of the reviewer, we also elaborated on a more detailed review on the plant responses to PAR and R:FR ratio in the introduction.

“Line 156-158: It seems to me that the information plants use to adjust internode lengths, height, and SRL in response to R:FR and light intensity are almost completely understood, and have been since the 1980s or even earlier. Again, this study shows the complexity of plant responses, and casts all of these exceptionally well understood responses in an ecological competition light, which is cool. I just don’t understand why the paper is written as if everything is novel. Your data are exciting! It seems to me you could just let them stand on their own and our field would be better served by your important contribution.”

We agree with the reviewer that the information used by plants to perceive competition and adjust their phenotype accordingly is already understood. However, in this case we aimed at explaining that plants not only respond to competition cues but that they can also respond to vertical gradients of these cues. We have now tried to better clarify this, particularly by referring to the newly-measured R:FR levels, as suggested by reviewer 1 and 2 (see below) (L173-181).

“METHODS: Line 235: I assume this plant is perennial? Please state this somewhere. Short studies of very long lived perennials are perfectly valuable, but these data only show short term responses during the establishment of very young ramets. Readers should know that the lifetime responses of long lived plants might be very different if this plant were allowed to live out its life (which could be decades or centuries long).”

According to this valuable comment we now stated that *P. reptans* is indeed a perennial plant (L249).

“Line 310: Given that PAR varied 5-fold between treatments; I’d be curious to see the GLMMs with PAR as a co-variate. Detangling the effects of R:FR from PAR intensity would greatly strengthen the interpretation of results. The expectation would be that low PAR produces shade leaves, and that high R:FR produces avoidance responses. But perhaps they interact! This would be exciting!”

We agree with the reviewer that correlating between the studied traits and PAR levels at the pots might have added valuable information. However, as these measurements were conducted in only a subset of the pots and not in all pots, as was erroneously mentioned in the previous version of the manuscript, we were unable to perform such a correlation.

We also agree with the second comment, that disentangling the effects of PAR and R:FR could be valuable as well. We thank the reviewer for this comment, which also coincides with a similar suggestion of reviewer 2 to add information of R:FR. Due this suggestion we have now added actual measurements of R:FR at the different heights within the treatments. These results are now shown together with the PAR measurements across heights, which were previously shown in the supplementary information (now in Fig. 2). Due to this addition we removed the figure that reported differences in PAR levels between treatments at initial plant height (previously shown in Fig. 1). The results of the R:FR measurements suggest that it increases significantly with height only within the short-dense treatment (Fig. 2b). As suggested by the reviewer, this result greatly strengthens our interpretation of the results, showing that indeed only this R:FR gradient induced increased vertical inclination. This added information is now referred to also in the discussion (L173-181).

“Line 311: If the experiment was organised as a random block design, why isn’t block a random effect in the GLMMs?”

As plants were arranged in blocks according to genotype (L283-284), we used “genotype” as a random “blocking” factor

Reviewer 2

*“The manuscript argues that 1) the shading setup produces complex light cues that mimic different competitor heights and densities found in nature, 2) *Potentilla reptans* produces three very different responses to competitors in response to these treatments, and 3) these responses are likely adaptive to natural competition.*

Evaluation of these three arguments.

1. Argument 1) I agree that the experimental setup, by using a shade filter that lowers R:FR and creating strips that mimic plant stems, is appropriate methodology for mimicking different types of competition. The authors should report how R:FR differs among treatments, as well as irradiance, since low R:FR is the predominant light cue of competitors.”

We thank the reviewer for this valuable comment, due to which we now performed additional measurements using the original experimental setup. Namely, we added measurements of R:FR at the different heights within the treatments. These results are now shown together with the PAR measurements across heights, which were previously shown in the supplementary information (Fig. 2). The results of these measurements greatly strengthen our initial interpretation and suggest that while PAR availability increases with height in the two short treatments (Fig 2a), R:FR increases significantly only in the short-dense treatment (Fig. 2b). Thus, we could demonstrate that each of the four competition treatments provides an exclusive light environment which is characterized by a unique light intensity, and/or light quality and/or gradient in these features, which can be perceived by a plant. This result provides a more in-depth explanation for the greater vertical inclination exhibited mainly under the latter treatment, suggesting that only when neighboring vegetation is dense but surmountable, plants experience as they grow, a gradient of improved R:FR as the main competition cue (see discussion, L173-181).

“2. Argument 1) Note that while the setup in using strips of filter and using 2 cylinders of strips provides a more realistic simulation of light competition than would the same combinations of irradiance and R:FR created by overhead filter sheets (the most common way to simulate competition), they have not provided experimental evidence or references that indicate the realism makes a difference.”

We thank the reviewer for this comment due to which we now added this clarification and added reference to previous studies to support it (L96-97).

“3. Argument 3) I agree that the adaptive interpretation of the three behaviours is consistent with the current consensus given functional arguments, comparisons of how population and species from different competitive environments respond to low R:FR and irradiance, and measures of natural selection for vertical growth.

4. Argument 3) The authors could better explain the traits and their function to an audience unfamiliar with these traits - see below.”

We thank the reviewer for this comment and, as specified below, have added explanations as well as a figure to clarify the measured traits.

“5. Argument 2) I have concerns with the data interpretation related to the claim of responses specific to each type of competition. For petiole length and stolon length, the interpretations do match with the statistics in Table 1 and means in Fig 2. However, for height-per-diameter and specific leaf area, the statistics do not support the interpretation. The wording of the manuscript implies the evidence for three different competitive responses is that the responses depend on the combination of filter density and height. Such a specific response would be indicated by a significant Height X Density interaction in the analyses of variance. While the means for height-per-diameter and specific leaf area reported in in Figure 2 follow the patterns expected from Height X Density interactions, the statistical analysis shows that the Height X Density interactions are in fact not significant.

*a. "Unlike plant petiole length and height, the vertical inclination exhibited by *P. reptans*, which was measured as height-per-diameter ratio, was greatest when plants were subjected to simulated short-dense neighbours (Table 1, Fig. 2c, Fig. 3; ". Here, Table 1 indicates significant responses in height per diameter to Height and Density, but no significant HeightXDensity interaction. Fig 2c indicates that height per diameter is greatest in short dense, and differs significantly from tall sparse and tall dense, but not short sparse.*

*b. "In contrast to their vertical inclination, *P. reptans* exhibited the highest shade tolerance, which was measured as leaf area and specific leaf area, when subjected to simulated dense and tall neighbours (Table 1, Fig. 2d, Fig. 3)" Here Table 1 indicates a significant effect of Height and a nearly significant Height X Density interaction for specific leaf area. Figure 2d indicates that the specific leaf area for tall dense differs significantly from that of short sparse and dense, but not from tall sparse. Figure 3 indicates that specific leaf area and leaf area responses are highly correlated.*

6. Argument 2) This mismatch between how the treatments appear to differ and whether the statistics back up that appearance needs to be resolved such that the interpretation follows the statistical analysis. Here are possible approaches, with the first two related to further statistics, and the second two at changing the interpretation of the present statistics: 1) It would be useful, if not already done, to check that the residuals meet the assumptions of anova, particularly for homoscedasticity. 2) The partial redundancy analysis is interesting, but purely descriptive. However, the authors could look for the interactions in multivariate space. 3) The interpretation could be based on the significant main effects only. 4) The interpretation could note that the main effect only is significant, and that the main effect is clearly resulting from the larger mean in a particular treatment combination.”

We thank the reviewer for this comment due to which we now rephrased the wording in the results to better clarify our interpretation of them. In particular, for height-per-diameter we do not suggest a significant interaction between height and density, but rather that both factors had a significant effect, as suggested by the reviewer, which resulted in having the highest values under the short-dense treatment (L122-123). In this case, even if density would cause the same increase within the two height treatments (i.e. no interaction), the fact that height also had a significant effect would still result in having one particular treatment with the highest values.

For specific leaf area however, only filter height had a significant effect, but the highest values were exhibited under the tall-dense treatment as revealed by the mean comparisons.

Here, we now suggest more clearly, as also noted by the reviewer, that the higher values under the tall-dense treatment could be reflected in the nearly significant effect of the height x density interaction (L133-134).

Moreover, we have now performed post-hoc mean comparisons with the slightly less strict false discovery rate correction (Benjamini and Hochberg, 1995 *Journal of the Royal Statistical Society* 57: 289-300), instead of the overly conservative Bonferroni correction, as has been recommended for ecological studies by Verhoeven et al. 2005 (*OIKOS* 108: 643-647). The results of these mean comparisons indicate that height-per diameter was significantly higher in the short-dense treatment compared to all other treatments (including the short-sparse one) (Fig. 2c). Similarly, specific leaf area was significantly higher in the tall-dense treatment compared to all other treatments (Fig. 2d). We believe that these results, using the less strict but still conservative multiple-comparison correction (Noble 2009, *Nature Biotechnology*), can provide greater support of our claim of single response strategies per each competition scenario and could be particularly valuable in preventing future confusion by readers as to our interpretation of the results.

Furthermore, as suggested by the reviewer, in the multivariate space we performed simple tests for each treatment with a false discovery rate correction for multiple tests. This analysis revealed a significant effect of each of the treatments on the plastic responses of *P. reptans* within the multivariate space (Fig. 4 legend), i.e. our results were confirmed by these additional analyses.

We would also respectfully disagree with the reviewer that the partial redundancy analysis is purely descriptive (as is for example an unconstrained analysis, such as principal component analysis). Rather, the partial redundancy analysis is a multivariate analog to a univariate GLMM: the pseudo-F statistic in a constrained analysis has a similar meaning as the F-ratio in ANOVA of the regression model and the Monte Carlo permutation test is used in an analogous way. It works with the null hypothesis stating the independence of the response data and the explanatory variables, in our case after filtering out the effect of covariates (Lepš and Šmilauer 2013 – *Multivariate Analyses of Ecological Data using Canoco* 5). Therefore, the significance (overall, of both constrained axes and of the simple effects) shows that the particular plastic responses of *P. reptans* to the simulated neighbors' treatments are different from what we would expect by chance. However, we apologize if we misunderstood the reviewer's suggestion and will be happy to provide further analyses to strengthen our results and their interpretation.

“1. The paper structure can be tightened up and made more concise, less review-style. In particular, the first paragraph of paper is short and weak. Foreshadow the direction of the paper - that there are different kinds of responses to competitor - tolerate, evade, avoid, resist, escape. The question is how precisely can plants tailor their response to the kind of competitor present. Is it simply genetic differentiation in which traits are cued by low R:FR, or can plants sense more than low light and low R:FR indicating a mass of chlorophyll and respond adaptively to complex cues from light.”

We thank the reviewer for this comment and have accordingly removed the first paragraph. Instead, we begin the introduction focusing on the different plastic responses plants can exhibit in response to light competition.

“2. Terminology: Rather than calling it a shift, call it phenotypic plasticity, developmental plasticity, response to the environment. Be specific that the experiment examines looking at responses to light cues.”

Although this experiment did study the plastic responses of plants to light competition, we did not examine plastic responses per se, but rather the ability of plants to choose (or shift) between different categories of plastic responses. We used the term “shift” following Novoplasnsky 2009 (Plant, Cell & Environment 32: 726-741), who originally suggested this idea, and therefore feel that it is more appropriate here than mere phenotypic plasticity.

“3. Terminology: Petiole length is not synonymous with petiole elongation. While greater petiole elongation causes longer petioles, but larger leaves and older leaves will have longer petioles. To be a measure of elongation, the length must be relativized to either lamina size (mass or area), aboveground mass, petiole mass, or other measure of size. For the current trait, use petiole length throughout.”

According to this comment we changed the term “petiole elongation” to “petiole length” in all text pertaining to our results.

“4. Terminology: Authors need to be consistent in discussing vertical growth - the same trait is variously identified as vertical inclination and height per diameter. Petiole length is sometimes included as a component of vertical growth, but sometimes not. Vertical inclination itself needs defining as a term.”

In accordance with this comment, we changed our reference to petiole length as part of the vertical inclination response. We agree with the reviewer that the term vertical inclination is also identified as height-per-diameter in the figures and tables, but we believe that using the latter term there is more appropriate, because this was the actual measured trait. We therefore added this term in brackets when vertical inclination is referred to in the results and in the first time it is referred to in the discussion. In addition, we now refer to vertical inclination in the introduction (L24-25), as well as provide an explanation in the introduction for the way this trait was measured in our study (L101).

“5. Missing reference: Crepy, M. A., and J. J. Casal. 2014. Photoreceptor-mediated kin recognition in plants. New Phytologist: on complex light cues that indicate identity.”

This study is now referred to in L30 and 79.

“6. Missing reference: Huber, H., A. Fijan, and H. J. During. 1998. A Comparative Study of Spacer Plasticity in Erect and Stoloniferous Herbs. Oikos 81:576-586 on vertical vs. horizontal spacers.”

This study is now referred to in L221

“7. A plant diagram is needed that identifies the traits measured. Most people are not familiar with rosette plant traits.”

According to this comment we added a diagram of some of the measured traits, including height, diameter, petiole length and internode length (Fig. 1g).

“8. The introduction should also focus on the traits measured. For example, the paragraph on vertical growth focuses on stem elongation and shade avoidance. While petiole elongation is measured, there is no discussion of vertical inclination/petiole angle.”

As mentioned above, we now refer in the introduction to vertical inclination as a vertical shade-avoidance response (L24-25), and to internode length as a lateral shade avoidance response of clonal plants (L52).

“9. Figures 1,3,and 4 are difficult to read in black and white. Many symbols and lines are nearly or completely invisible.”

According to this comment we have now tried to increase the color contrast in the figures, and added different symbols (Fig. 5), which we hope helps their appearance in black and white. In accordance with the suggestions from reviewer 1 and 2, we now use colorblind safe and print friendly colors. We also verified the choice by a color blind colleague and by printing the figures in black and white. We have also added different symbols to figures 2, 4 and 5 and we use the same color and symbol code throughout all figures to facilitate their understanding.

Reviewer 3

“This paper claims to show that plants show distinct strategies in response to vegetation height: shade tolerance in tall vegetation and competitive shade avoidance in short vegetation. Proxies of these two strategies are increases in SLA and in length/diameter, respectively. My main objection to this set up is that increases in SLA are not really a shade tolerance strategy. Across species shade tolerant species typically exhibit lower SLA and less shade-induced plasticity than light demanding species. This makes sense as high SLA is associated with less durable leaves and relatively greater chances of leaf loss through damage that is detrimental in the shade.”

Although we agree with the reviewer that some shade tolerant species exhibit lower SLA and less shade-induced plasticity, *P. reptans* is not such a shade-tolerant plant, but rather occurs in meadows or disturbed environments. Previous studies have also shown that SLA in *P. reptans* increases in response to shade and light competition cues (e.g. Stuefer and Huber 1998, *Oecologia*: 117: 1-8), and even more studies have demonstrated that SLA is a very common shade tolerance response in many species (e.g., reviewed by Gommers et al., 2013 *Trends in Plant Science* 18: 65-71. We therefore believe that our use of this trait is valid as a shade tolerance response.

“Furthermore I find the result for the shift in the relation between petiole length and diameter with a $P = 0.049$ rather weak. To confirm whether such low significance is really consistent I would want to see a second experiment.”

We were not certain as to which result the reviewer referred to in this case, because we did not examine a shift between petiole length and diameter in our study. The reviewer might refer here to the relationship between plant height and diameter, and in particular to the differences in elevation between treatments in the standardized major axis analysis. We agree that a P value of 0.049 is rather close to the commonly used threshold of 5%, but suggest that this result only refers only to the effect of filter height, and should be evaluated together with the shift along a common slope, which refers to the effect of density and which was also highly significant ($p < 0.001$), as well as with the GLM results, which show significant effects of both height and density on height-per-diameter. Therefore, even if that result would not be statistically significant, our overall conclusions would remain the same.

REVIEWERS' COMMENTS:

Reviewer #1 (Remarks to the Author):

I have reviewed this manuscript once before. My main concern was that quite a lot is known about shade-avoidance in plants, but the authors wrote the first draft as if everything they found was novel.

I think the authors have addressed all of my concerns in this revised manuscript quite well. I appreciate that they were willing to take the time to re-frame the manuscript, and the result is something that connects their important results to the existing literature much more clearly. I also appreciate that they adjusted some of their figures and analyses as requested.

My only concern now is that the manuscript has quite a few typos throughout, and could be edited to be much more concise. I suspect the introduction and discussion could be reduced in length by 20-30% without any loss of information with some careful edits. The results and methods are quite clear already.

Some minor comments:

Example typos: Line 23 - Suite not suit? Line 26 - leaves not leave?

Line 60: I don't think there are too many examples where elongation is maladaptive. This is why it is a nearly universal response in plants. In under-stories, herbaceous plants still benefit from being taller than other herbaceous plants, and they still lose by being shaded twice by the canopy and by their herbaceous neighbours. Additionally, it's not so much that height is maladaptive to the individual plant in a crop field, it's that allocation is a zero sum game. So, while natural selection shapes wild plants to find a height that maximises the trade-off between photosynthesis and competition, agronomists use natural selection to minimize competition and maximize photosynthesis. In other words, humans change the rules of competition in crop fields. This is a pretty important fact, as interpreting crop growth which was shaped by artificial selection through a lens of natural selection can lead to some pretty strange conclusions.

Reviewer #2 (Remarks to the Author):

I am satisfied with the revisions. I see this as a paper that will be broadly interesting to botanists, ecologists, and evolutionary biologists.

Reviewer #3 (Remarks to the Author):

The authors have nicely dealt with all comments

Reviewer 1

“My only concern now is that the manuscript has quite a few typos throughout, and could be edited to be much more concise. I suspect the introduction and discussion could be reduced in length by 20-30% without any loss of information with some careful edits.”

In accordance with this comment, we attempted to reduce the length of the introduction and discussion. However, we believe that a 30% reduction could not be feasible without loss of important information in these sections, and therefore could reduce the length of the introduction and discussion by 20 and 10% respectively. We hope this reduction is satisfactory.

“Example typos: Line 23 - Suite not suit? Line 26 - leaves not leave?”

These typos were corrected and the manuscript was checked for additional ones.

“Line 60: I don't think there are too many examples where elongation is maladaptive. This is why it is a nearly universal response in plants. In under-stories, herbaceous plants still benefit from being taller than other herbaceous plants, and they still lose by being shaded twice by the canopy and by their herbaceous neighbours. Additionally, it's not so much that height is maladaptive to the individual plant in a crop field, it's that allocation is a zero sum game. So, while natural selection shapes wild plants to find a height that maximise the trade-off between photosynthesis and competition, agronomists use natural selection to minimize competition and maximize photosynthesis. In other words, humans change the rules of competition in crop fields. This is a pretty important fact, as interpreting crop growth which was shaped by artificial selection through a lens of natural selection can lead to some pretty strange conclusions.”

We thank the reviewer for this clarification and have accordingly removed the reference to crop fields in this explanation as well as refrain from referring to vertical elongation as ‘maladaptive’.